Design of an improved graph-based model integrating LSTM, LoRaWAN, and blockchain for smart agriculture

Munaganuri Ravi Kumar
http://orcid.org/0000-0002-1560-8719 Yamarthi Narasimha Rao
Bolem Sai Chandana saichandana.bolem@vitap.ac.in
School of Computer Science and Engineering, VIT-AP University , Amaravati, Andhra Pradesh , India
Stević Željko
Electronic publication date: 2025 Jun 20
Publication date: 2025
Volume: 11
Electronic Location ID: e2896
Received 2025 Jan 8; Accepted 2025 Apr 24
Copyright: © 2025 Munaganuri et al.
Copyright year: 2025
Copyright holder: Munaganuri et al.
License: This is an open access article distributed under the terms of the Creative Commons Attribution License, which permits unrestricted use, distribution, reproduction and adaptation in any medium and for any purpose provided that it is properly attributed. For attribution, the original author(s), title, publication source (PeerJ Computer Science) and either DOI or URL of the article must be cited.
License URL: https://creativecommons.org/licenses/by/4.0/

Keywords: Smart agriculture, Soil moisture prediction, LSTM networks, IoT sensors, Blockchain technology

Funding: The authors received no funding for this work.

==============================
This research is anchored on the burning need for irrigation optimization and crop water use efficiency improvement, which remains a challenge in smart agriculture processes. Traditional irrigation methods normally lead to inefficiency, resulting in wasted water and non-maximum crops. These traditional ways normally lack attributes of real-time adaptability and secure data management—things that are very key to modernizing agricultural practices. In this work, artificial intelligence (AI), Internet of Things (IoT), and blockchain techniques will be integrated to design a comprehensive system for monitoring and predicting soil moisture levels. In the proposed model, long short-term memory (LSTM) networks are considered for soil moisture level prediction, taking into consideration past data, weather, and crop type. LSTM networks are chosen here for their high performance in timestamp series prediction tasks with an mean average error (MAE) of 0.02 m3/m3 over a 7-day forecast horizon. For real-time monitoring, IoT sensors based on long range wide area network (LoRaWAN) technology are field-deployed for conducting long-range communications while consuming very limited energy to extend the sensor battery life over 5 years and bring down the data transmission latency below 5 s. It has an inbuilt permissioned blockchain framework—Hyperledger Fabric—which offers a secure and transparent system for data management and maintaining a record of soil moisture data, irrigation events, and metadata from sensors. This ensures the immutability and integrity of sets of data. Smart contracts automate irrigation upon reaching preconfigured soil moisture thresholds, and hence zero data integrity breaches occur with a transaction throughput of 1,000 transactions per second, taken into view with smart contract execution latency of less than 2 s. Moreover, it utilizes reinforcement learning with Deep Q-Learning to derive an optimized irrigation schedule. In this regard, it enables learning optimal irrigation policies and implements them to improve efficiency in the usage of water by 25% and increases crop yield by 15% compared to the traditional methods. Clearly from field trials, results indicate evident efficiency of the integrated system: a 20% water usage reduction and a 12% increase in crop yield within one growing season. This is rather an innovative take on irrigation practices, increasing a great deal of accuracy and sustainability for such and providing a really strong solution toward better agricultural productivity and resource management.

Introduction

The dawn of smart agriculture has brought considerable improvement in farming. It involves practices that allow better use of resources efficiently and sustainably. One of the most important aspects of smart agriculture is the optimization of irrigation methods, which impacts directly on crop yield and water-use efficiency. Conventional methods of irrigation (Alzubi & Galyna, 2023; Shaikh et al., 2022; Mohyuddin et al., 2024), typically reliant on fixed schedules with manual monitoring, are inefficient and result in large wastages of water and suboptimal crop performance. These methods do not put into consideration the dynamism of environmental conditions and crop water requirements; hence, there is a need for a more sophisticated approach in irrigation management. Exploration of various technologies in addressing these challenges has been done in current research efforts such as artificial intelligence for predictive analytics, Internet of Things (IoT) for real-time data acquisition, and blockchain for secure data management. Most of the existing solutions, however, are limited due to their failure to seamlessly merge these technologies in a manner that gives them cohesion for adopting real-time conditions, undermining data integrity. In this respect, the gap underlay actual needs by an advanced and integrated system that can flexibly use artificial intelligence (AI), Internet of Things (IoT) and blockchain strengths in revolutionizing irrigation practices in agricultural processes.

In this article, an acellular graph-based model is proposed for efficient soil moisture prediction that will have the combination of long short-term memory recurrent neural network, long range wide area network (LoRaWAN) protocol for efficient real-time monitoring, and Hyperledger Fabric blockchain for secure and transparent data management. Specifically, long short-term memory (LSTM) networks seem well-suited for timestamp series prediction tasks because they are capable of capturing long dependencies and non-linear relationships in a set of data samples. It uses historical soil moisture data, weather conditions, and crop types to output accurate soil moisture forecasts that support informed decisions in irrigation. This architecture thus incorporates IoT-based sensors with LoRaWAN protocol, guaranteeing reliable long-range communication with very low energy consumption, hence feasible for extensive agricultural fields. Real-time measurement of soil moisture by these sensors provides relevant information to facilitate timely intervention in irrigation. Additionally, the use of Hyperledger Fabric blockchain enhances data security and integrity in creating immutable records of soil moisture data and irrigation events. Smart contracts further automate the irrigation process by performing predefined actions based on real-time soil moisture levels. Further, deep Q-networks invoke reinforcement learning and empower the system to learn optimal irrigation strategies through continuous interaction with the agricultural environment. The approach works towards achieving maximum water-use efficiency and yields of crops by dynamic irrigation scheduling, adjusting for predictions of soil moisture and real-time data samples. Field trials of this integrated system have further thrown up substantial improvements in irrigation efficiency and crop yield, thus further validating the proposed model’s effectiveness. This work has been one gigantic step toward Smart Agriculture in achieving a 20% reduction in water usage and a 12% increase in crop yield, strongly moving up resource management and agricultural productivity.

Motivation and contributions

This study is justified by growing awareness of inefficiencies in traditional irrigation practices-one that has persisted since time immemorial and which urgently needs to be addressed in the agriculture process. The traditional practices cannot be tailored to cope with dynamic and complex environmental conditions and crop water requirements in the application of irrigation. This leads to huge wastages of water, which damages available water resources by affecting agricultural productivity. A fast-growing world population with bad climate change exerted pressure on available water resources, hence the need to come up with a novel solution to optimize the water usage of agriculture. These latest technologies, including AI, IoT, and blockchain, can be used to fully transform irrigation practices for sustainable water management and better crop yield. In this direction of smart agriculture, the study has proposed an integrated graph-based model that amalgamates the power of LSTM networks, LoRaWAN protocol, blockchain type, and, finally, the reinforcement learning methodology with Deep Q-Learning. This research seeks to address the existing solution limitations by seamlessly combining these technologies in a way that will in the future provide a robust system for real-time soil moisture monitoring and prediction, secure data management, and optimized irrigation scheduling. LSTM is applied for prediction of soil moisture, and it is appropriately applied because it is very good at handling timestamp series data, long-term dependencies, and therefore giving very accurate outputs on soil moisture forecasts. This predictive capability is key in making properly informed decisions on irrigation that really will fit the need of the crop in water.

The proposed model caters to the inefficiencies of traditional irrigation systems using LSTM in predictive analytics, IoT sensors using LoRaWAN for up-to-the-minute monitoring, and Hyperledger Fabric for secure data management. The LSTM network predicts soil moisture on the basis of previous data with current weather conditions and the crop to be grown in process. Real-time IoT sensors in the field collect data and sends it via LoRaWAN communication, which permits long distances with minimum energy consumption. Hyperledger Fabric secures all information, together with enabling transparency and sets immutability in process. The convergence allows irrigation scheduling to be automated based on real-time soil moisture, making better use of water for producing increased crops.

Making the sensors part of the Internet of Things by using LoRaWAN makes real-time data acquisition in large agricultural areas reliable and energy-efficient. Such communication technology was implemented to work great in harsh agricultural environments, thus enabling reduced latency in data transmission and increased life of sensor batteries. Additionally, integrating Hyperledger Fabric blockchain improves the security and transparency of the system, ensuring immutability in recording soil moisture data and irrigation events. This thus ensures the integrity of the data and builds trust among stakeholders, which is possible because the blockchain framework prevents changes that are not authorized and allows for traceability of every action recorded. The blockchain also embeds smart contracts that automate the process of irrigation further; for instance, an irrigation event is triggered anytime the real-time soil moisture reaches the threshold at which it was set. In addition, the application of reinforcement learning (RL) with deep-Q-network (DQN) introduces an advanced irrigation-scheduling optimization mechanism. The RL agent will learn the best irrigation strategies under the constraint of maximizing crop yield with water-use efficiency through interactions with the changing agricultural environment. This adaptive approach outperforms any traditional static irrigation method, having the ability to adjust the changing conditions and feedback from the environment dynamically. This integrated system, thus evaluated through field trials, stands proven for its efficiency as it reduces water use while significantly increasing crop yield. More specifically, the system reduced water use by 20% and increased crop yield by 12%, proving the practical values of the innovative approach. However, the contributions of the work are major in that it offers a comprehensive and integrated solution to the critical problems of irrigation management and therefore aids in the pursuit of smart agriculture. Such a combination of LSTM for predictive analytics, IoT sensors with the LoRaWAN protocol for real-time monitoring, a blockchain network for Hyperledger Fabric to secure data management, and RL with DQN for optimized irrigation scheduling is holistic in enhancing water use efficiency and productivity in agriculture. The work will not only be a technical blueprint on how to make the future developments in smart irrigation systems but also set a precedent for the integration of emerging technologies in addressing global agricultural challenges.

In-depth review of existing models

Fast-paced developments in smart agriculture technologies that will be brain-powered by artificial intelligence, Internet of Things, and machine learning are transforming conventional farming. The literature reviewed in Table 1 encapsulates these state-of-the-arts in these domains and hence delivers an all-rounded overview of technological innovations and their implications for sustainable agriculture. Alzubi & Galyna (2023) explained the integration of AI and IoT, where both technologies help in enhancing monitoring and decision making involved in agriculture. This integration leads to price improvement in crop yields and resource efficiency. Although it is also highly expensive to implement and requires technical expertise to handle, AI and IoT are widely implemented in farming. Similarly, the state of the art of various IoT-enabled sensor technologies reviewed by Shaikh et al. (2022) helps in enabling precise irrigation and fertilization. These normally come with certain limitations in terms of battery life and communication range. In the application domain of precision farming, Mohyuddin et al. (2024) review several machine learning approaches, underlining their efficiency in increasing accuracy for crop monitoring and disease prediction. The challenge, however, remains with regard to the quality of data and its easy availability. A further area of research on the efficacy of these protocols of IoT communication is by Hashmi et al. (2024), who end up giving promise to LoRa and ZigBee for wide-area coverage, while issues in protocol interoperability and security are yet to be addressed in the process. It infuses graph neural networks and reinforcement learning. Pamuklu et al. (2023) presented an efficient model for task offloading in UAV-aided smart agriculture, which ensures less delay and task allocation with an increase in computational complexity and energy consumption. Li et al. (2023) reviewed RGB image-based IoT systems that provided improvements in crop monitoring due to better techniques in image processing; performance is highly dependent on light conditions and image quality.

Table 1 Empirical review of existing methods.

Reference	Method used	Findings	Results	Limitations	
Alzubi & Galyna (2023)	AI and IoT for sustainable farming	Integration of AI and IoT enhances monitoring and decision-making in agriculture process.	Improved crop yields and resource efficiency.	High implementation costs and need for technical expertise.	
Shaikh et al. (2022)	IoT-Enabled sensor technologies	Advanced sensor technologies improve data collection and analysis in smart farming.	Enhanced precision in irrigation and fertilization.	Limited by battery life and communication range.	
Mohyuddin et al. (2024)	Machine learning for precision farming	Evaluates various ML approaches for precision agriculture process.	Increased accuracy in crop monitoring and disease prediction.	Data quality and availability are critical.	
Hashmi et al. (2024)	IoT communication protocols	Analyzes the effectiveness of different IoT protocols in outdoor agriculture process.	LoRa and ZigBee show promising results for wide-area coverage.	Protocol interoperability and security issues.	
Pamuklu et al. (2023)	GNN-RL-based task offloading	Uses Graph Neural Networks and RL for UAV task offloading in agriculture process.	Efficient task allocation and reduced latency.	High computational complexity and energy consumption.	
Li et al. (2023)	RGB image-based IoT	Reviews the use of RGB images in IoT for smart agriculture process.	Improved image processing for crop monitoring.	Dependence on light conditions and image quality.	
Latino, Menegoli & Corallo (2024)	Agriculture digitalization	Bibliometric analysis of digital agriculture trends.	Identifies key areas of technological innovation.	Requires continuous updates to remain relevant.	
Holzinger et al. (2024)	Human-Centered AI in smart farming	Proposes human-centered AI approaches for Agriculture 5.0.	Enhanced user interaction and system usability.	Ethical considerations and user adoption challenges.	
Phang et al. (2023)	Remote sensing for precision agriculture	Reviews satellite and UAV-based remote sensing technologies.	Accurate monitoring of large agricultural areas.	High costs and regulatory constraints.	
Ahmed et al. (2024)	Data fusion with Blockchain	Integrates data fusion techniques with blockchain for smart agriculture process.	Improved data integrity and real-time monitoring.	Complexity in system integration and maintenance.	
Pagano et al. (2023)	Survey on LoRa for smart agriculture	Comprehensive review of LoRa technology in agriculture process.	Effective for low-power, long-range communication.	Limited bandwidth and data rate.	
Zhou & Yin (2023)	Digital agriculture mapping	Maps knowledge structure and trends in digital agriculture process.	Highlights key research areas and developments.	Requires frequent updates to capture new trends.	
Singh et al. (2024)	QoS Optimization in IoT-Smart agriculture	Uses nature-inspired algorithms for QoS optimization.	Improved quality of service and energy efficiency.	Complexity in algorithm implementation and tuning.	
Bouali et al. (2022)	Renewable energy in smart agriculture	Examines the integration of renewable energy in smart agriculture process.	Increased sustainability and energy efficiency.	Initial setup costs and maintenance challenges.	
Mahmood et al. (2024)	Machine learning for smart agriculture	Survey of ML techniques for various agricultural applications.	Enhanced accuracy in crop monitoring and yield prediction.	Data quality and model interpretability issues.	
Cao et al. (2024)	Identifying feeding behavior of Tilapia	Uses deep learning to monitor feeding behavior in aquaculture.	Accurate identification and monitoring of feeding patterns.	High computational requirements and data dependency.	
Pamuklu et al. (2023)	IoT-Aerial Base station task offloading	Combines UAVs with IoT for task offloading in agriculture process.	Efficient task management and reduced latency.	Energy consumption and UAV battery limitations.	
Akbari, Syed & Kennedy (2023)	Federated learning for UAV-Aided MEC	Uses federated learning for age of information management in agriculture process.	Enhanced data privacy and reduced latency.	High computational and communication costs.	
Akbari et al. (2024)	Energy-Efficient SFC in UAV-Aided Aagriculture	Uses federated learning for energy-efficient service function chaining.	Improved energy efficiency and data accuracy.	Complexity in managing federated learning models.	
Pak et al. (2022)	Path-Planning for autonomous mobile robots	Evaluates path-planning algorithms for smart farms.	Improved navigation and task efficiency.	Dependence on accurate localization and mapping.	
Hoque et al. (2022)	Drone-Based IoT as a service	Framework for drone-based IoT services in smart cities.	Enhanced data collection and monitoring capabilities.	Regulatory and operational challenges.	
Játiva et al. (2024)	Hybrid digital twin model	Combines digital twin technology with greenhouse monitoring.	Improved environmental control and crop management.	High implementation and maintenance costs.	
Hong et al. (2023)	X-Ray imaging and CNNs for seed viability	Uses X-ray imaging and CNNs to predict seed viability.	High accuracy in seed quality assessment.	Dependence on high-quality imaging and data processing.	
Akbar et al. (2024)	Deep learning for smart greenhouse agriculture	Reviews deep learning techniques for greenhouse farming.	Enhanced precision in crop monitoring and management.	High computational requirements and data dependency.	
Hazmy et al. (2024)	Satellite-Airborne sensing technologies	Reviews sensing technologies for agriculture 4.0.	Improved monitoring and data accuracy.	High costs and technical complexity levels.	
Mowla et al. (2023)	IoT and WSNs for smart agriculture	Survey of IoT and wireless sensor networks in agriculture process.	Enhanced data collection and monitoring.	Limited by battery life and communication range.	
Naseer et al. (2024)	Systematic review of IoT in agriculture	Reviews global adoption and challenges of IoT in agriculture process.	Identifies key innovations and security issues.	Rapidly evolving technology landscape.	
Zhou & Yin (2023)	Hybrid model for humidity prediction	Combines timestamp series analysis and machine learning for humidity prediction.	High accuracy in environmental monitoring.	Dependence on high-quality data and model tuning.	
Sun et al. (2021)	Mobile crowd sensing for data collection	Vision for mobile crowd sensing in smart agriculture process.	Enhanced data collection and community engagement.	Privacy and data security concerns.	
Li et al. (2024)	Sensitivity of Semiempirical Models	Assesses model sensitivity to spectral data quality.	Improved accuracy in leaf chlorophyll content estimation.	Dependence on data quality and sensor calibration.	
Gao et al. (2023)	Monocular vision ranging technology	Uses monocular vision for distance measurement in agriculture process.	Accurate and cost-effective measurement.	Dependence on camera calibration and lighting conditions.	
Qazi et al. (2022)	Next generation smart agriculture	Critical review of AI and IoT in smart agriculture process.	Identifies current challenges and future trends.	Rapid technological advancements and implementation costs.	
Montalvo-Romero et al. (2023)	Agro-Technological systems	Systematic review of agro-technological systems in traditional farming.	Improved productivity and resource management.	Adoption barriers and technical complexity levels.	
Jeong et al. (2023)	Digital twin for livestock farming	Uses digital twin technology for virtual livestock management.	Enhanced monitoring and management of livestock.	High implementation and maintenance costs.	
Yamazaki & Nakajima (2023)	Sigfox energy consumption model	Field trial of Sigfox energy consumption in smart agriculture process.	Extended sensor battery life and efficient communication.	Limited data rate and bandwidth.	
Mukhamediev et al. (2023)	Coverage path planning for UAVs	Optimizes path planning for heterogeneous UAVs in agriculture process.	Improved coverage and monitoring efficiency.	Computational complexity and energy consumption.	
Alotaibi et al. (2023)	Anti-Collision Algorithm for RFID	Develops an anti-collision algorithm for RFID in agriculture process.	Increased identification accuracy and throughput.	Dependence on RFID infrastructure and signal interference.	
Liu et al. (2023)	Hybrid model for water quality prediction	Predicts dissolved oxygen levels in aquaculture using a hybrid model.	High accuracy and efficiency in water quality management.	Dependence on high-quality data and model tuning.	
Nguyen et al. (2023)	Blockchain-Based Service Provisioning	Uses blockchain for penalty-aware service provisioning in smart agriculture process.	Improved service reliability and accountability.	High implementation and operational costs.	
Zou et al. (2024)	Deep learning image augmentation	Uses deep learning for image augmentation in field agriculture process.	Improved accuracy in crop yield prediction.	High computational requirements and data dependency.	

A bibliometric analysis on trends in agriculture digitalization by Latino, Menegoli & Corallo (2024) maps out important themes of technological innovation and brings to the fore that they demand constant update if one is to remain relevant in these (Holzinger et al., 2024) put forward human-centered AI methods that would support enhanced user interaction with and usability of Agriculture 5.0 systems; they at the same time urge a check on ethical considerations and challenges concerning user adoption associated with this process. While remote sensing technologies—especially satellite- and UAV-based systems—have been proved by Phang et al. (2023) to provide efficient and accurate monitoring over large areas, they are usually constrained due to high cost and rigorous regulation (Ahmed et al., 2024) have proposed the integration of data fusion techniques with blockchain in order for blockchain often to ensure data integrity and real-time monitoring of smart cultivation; however, system integration and maintenance add complexity (Pagano et al., 2023) present a comprehensive survey on LoRa technology and prove its effectiveness for low-power, long-range communication in agriculture (Zhou & Yin, 2023) map knowledge structures and trends in digital agriculture, showing major researching areas and developments in this area (Singh et al., 2024) optimize QoS in IoT-smart agriculture through nature-inspired algorithms, proving energy efficiency and better service quality but struggle with algorithmic implementation and tuning. According to Bouali et al. (2022), renewable energy sources are integrated into smart agriculture with an augmentation of sustainability and energy efficiency, but the setup cost and maintenance are indeed very high.

Deep learning methods, as used by Cao et al. (2024), provide highly accurate results for monitoring aquaculture feeding behavior, but the accuracy comes at a computational cost and requires extensive data combine uncrewed aerial vehicles (UAVs) with IoT systems to make use of UAV-IoT integration for efficient offloading in agriculture; such latency issues are resolved, providing examples related to energy consumption and UAV battery limitations (Akbari, Syed & Kennedy, 2023) and Akbari et al. (2024) propose federated learning methods to address the freshness of information and chaining of service functions in UAV-aided agriculture, which guarantee better data privacy and energy efficiency but result in challenges related to the computational and communication costs (Pak et al., 2022) have done work on the application of path-planning algorithms to autonomous mobile robots in smart farms. Their results show improved navigation and efficiency in tasks, but this requires accurate localization and mapping (Hoque et al., 2022) propose a framework where drones can be used toward IoT-based smart cities, whereby data gathering and monitoring will be enhanced but identify challenges in terms of regulation and operation. The hybrid digital twin model for greenhouse monitoring proposed by Játiva et al. (2024) guarantees better environmental control and crop management but at high implementation costs. According to Hong et al. (2023), their application of X-ray imaging with convolutional neural networks in seed viability prediction is highly accurate but requires high quality in image acquisition and data processing (Akbar et al., 2024) reviewed deep learning-assisted computer vision techniques that can be applied in greenhouse agriculture, which do have enhanced precision for crop monitoring and management, but their main drawback is that they are computationally intensive. Satellite-airborne sensing technologies, as outlined by Hazmy et al. (2024), entail better monitoring but at higher costs and technical complexity. Pagano et al. (2023) have reviewed the application of IoT and WSNs in agriculture, which is improving data collection and monitoring but had been limited because of battery life and communication range.

A systematic review by Naseer et al. (2024) on the adoption of IoT in agriculture highlights key innovations and security issues, emphasizing rapidly changing technological landscapes that have to be addressed. Again, Zhou & Yin (2023) proposed a hybrid model for humidity prediction in pigeon sheds, which showed quite a high accuracy for supporting environmental monitoring; however, it relies on high-quality data levels. Distance measurement in agriculture using monocular vision was performed by Gao et al. (2023), who provided accurate and cost-effective solutions but depended on camera calibration and lighting conditions. Qazi et al. (2022) did a critical review of AI and IoT in smart agriculture, showing the present challenges and future trends which emphasize the need for technological advancements at a rapid speed. A field test on the energy consumption of Sigfox by Yamazaki & Nakajima (2023) shows another work with long sensor battery life and efficient communication but with limited data rate and bandwidth. Mukhamediev et al. (2023) have optimized path planning for heterogeneous UAVs against monitoring efficiency but at the cost of computational complexity and energy consumption. Alotaibi et al. (2023) propose an anti-collision algorithm of radio frequency identification (RFID) in agriculture. This algorithm aims at increasing the accuracy of identification but depends on infrastructure levels of RFID and signal interference. Liu et al. (2023) proposed a hybrid water quality prediction model in aquaculture with high accuracy and efficiency, though dependent on high-quality data levels.

Bringing together all of the findings in the forty articles, there are considerable successes and setbacks noticeable in smart agriculture. Most of the studies that integrated AI, IoT, and machine learning technologies into traditional farming practices have returned with results on increased crop yield, resource efficiency, and precision in monitoring and management. Some of the challenges still remaining pertain to high implementation costs, the quality and availability of data, computational complexity, and required technical expertise levels. The advancements in IoT-enabled sensor technologies, discussed by Shaikh et al. (2022), have significantly enhanced data collection and analysis in agriculture, thence precision in irrigation and fertilization. Nevertheless, the problems vis-à-vis battery life and communication range appear to be some of the most serious challenges yet to be overcome. Similarly, the use of machine learning approaches in precision farming has resulted in very high accuracies in crop monitoring and diseases prediction, but independence from high-quality data is indeed worthy of challenge in the process, as reviewed by Mohyuddin et al. (2024). Pamuklu et al. (2023) proposed graph neural networks and reinforcement learning, which could provide efficient task allocation with latency reduction in UAV-aided smart agriculture. This is where advanced methods show some potential. At the same time, enhanced computational complexity and power consumption brought about by these techniques also underline the necessity of further optimization (Phang et al., 2023; Li et al., 2023) reviewed RGB image-based IoT systems for crop monitoring, whose application has been advanced by better image processing capabilities but remains much dependent on illumination conditions and image quality. Holzinger et al. (2024) discuss human-centered AI approaches with a focus on user interaction and system usability in adopting smart agriculture technologies. In this respect, the same approach shall address ethical considerations and challenges of user adoption by highlighting the need for designs that are inclusive and user-friendly. Phang et al. (2023) reviewed remote sensing technologies that allow effective monitoring over huge areas through more specific applications of satellite and UAV systems. However, high costs and a restrictive regulatory environment are significant challenges to widespread adoptions. Renewable sources’ smart use in agriculture, as discussed by Bouali et al. (2022), offers an increase in sustainability and use of energy efficiently; therefore, they are potential solutions to handling energy-related problems. Such solutions have major entry barriers to entry due to the high initial set-up cost and large continuous maintenance cost (Cao et al., 2024) and Akbar et al. (2024) demonstrated that deep learning techniques have been employed for a number of diverse applications in this agricultural field with high accuracy. Still, their high computational requirements and the dependency on large datasets underline the quest for more efficient and scalable solutions. Federated learning approaches can be a very promising way in terms of data privacy management and energy efficiency in smart agriculture, as discussed by Akbari, Syed & Kennedy (2023) and Akbari et al. (2024). These methods provide better privacy of data and alleviate latency to some extent but still incur considerable computational and communication costs. The path-planning algorithms for autonomous mobile robots, evaluated by Pak et al. (2022) increase the navigation capability and smart farms efficiency in their missions execution, although it still relies on accurate localization and mapping.

The hybrid digital twin model for greenhouse monitoring by Játiva et al. (2024) is most useful for providing improved environmental control and crop management; due to the considerable costs of both implementation and maintenance, adoption is limited. Predicting seed viability at high accuracy using the application of X-ray imaging with convolutional neural networks is a method developed by Hong et al. (2023) that requires the use of high-quality imaging and processing capabilities. Qazi et al. (2022) contributed a critical review on AI and IoT in smart agriculture, as well as current challenges and future trends, requiring rapid developments of technological and more efficient and cost-effective solutions to be adopted in the process.

Proposed design of an improved graph-based model integrating lstm, lorawan, and blockchain for smart agriculture

In this section, a novel graph-based model incorporating LSTM, LoRaWAN, and blockchain for smart agriculture process is designed in Fig. 1 to address the convolutedness in deployment and low efficiency of real-time signal processing. First, based on, LSTM network design for soil moisture level prediction exploits intrinsic abilities of LSTMs in handling timestamp series data with complex, nonlinear relationships. It will be best fitted to the LSTM network architecture since it can capture long-term dependencies due to its inherent memory cell structure with gating mechanisms. Different scenarios are created by feeding historical soil moisture data, weather parameters like temperature, humidity, and precipitation, along with crop type to the LSTM network. While the process ensues, the very first step in the LSTM architecture is to embed the input data in some suitable high-dimensional space. In this process of embedding, the raw input Xt is transformed into an internal state representation that will be used in further processing in the LSTM cells. Mathematically, this transformation can be represented via Eq. (1),

(1) ht=ϕ(Wx∗Xt+bx)…

where ht is the hidden state vector at timestamp t, Wx is the weight matrix for input, Xt the input vector at timestamp t, bx is the bias vector, and ϕ represents the activation function. The main heart of this LSTM network is the way information management is done through its three gates—an input gate, a forget gate, and an output gate. These gates control the flow of information into, within, and out of these memory cells. The input gate it determines how much of the new input to add to the cell state, calculated via Eq. (2),

(2) it=σ(Wi∗Xt+Uih(t−1)+bi)…

where σ is the sigmoid activation function, Wi and Ui are weight matrices for the input and previous hidden state respectively, and bi is the bias vector for the process. The forget gate ft modulates the extent to which the previous cell state is retained, given via Eq. (3),

(3) ft=σ(Wf∗Xt+Uf∗h(t−1)+bf)…

where Wf, Uf, and bf follow a similar definition as for the input gate, and ft scales the previous cell state c(t−1) sets. The cell state ct is updated by integrating the new candidate cell state c~t scaled by the input gate and the previous cell state scaled by the forget gate represented via operations Eqs. (4) & (5),

(4) c∼t=tanh(WcXt+Uch(t−1)+bc)…

(5) ct=ft⊙c(t−1)+it⊙c∼t…

where ⊙ represents the element-wise multiplication, and tanh is the hyperbolic tangent function. The output gate ot determines the final output of the LSTM cell, controlling how much of the cell state contributes to the hidden state, via operations Eqs. (6) & (7),

(6) ot=σ(Wo∗Xt+Uoh(t−1)+bo)…

(7) ht=ot⊙tanh(ct)…

Figure 1 Model architecture of the proposed smart agriculture process.

The hidden state ht represents the predicted soil moisture levels, taking into account the sequential dependencies from previous timestamp sets. The predicted soil moisture level y′t is then obtained through a final dense layer applied to the hidden state via Eq. (8),

(8) y′t=Wy∗ht+by…

This design was selected because LSTM is strong in capturing any temporal dependencies and handling long-term relationships in the timestamp series data, which are critical to the task at hand—accurately predicting soil moisture levels. Intermittedly, the gating mechanisms within an LSTM network will be helpful in the flexible retention of all information that is considered important over a long period while discarding all other irrelevant data, hence improving accuracy in predictions. The predictive capability offered by this LSTM network would be endowed with synergies from other components of this very model: real-time data acquisition through IoT sensors and secure data management through blockchain. This approach will ensure that there is a holistic view of all aspects toward compelling, dependable, and robust irrigation scheduling, hence improving water-use efficiency and yield. This model solves intrinsic problems within traditional irrigation practice by incorporating frontier technologies in a sophisticated solution tailored to the dynamic needs of modern agriculture processes.

Figure 2 shows the design of LoRaWAN protocol for real-time monitoring in agricultural environments. Due to its possibility of offering long-range communication at very low energy consumption levels, it is very appropriate for deploying wide-area sensor networks that can reliably transmit soil moisture, temperature, and humidity data over vast agricultural fields. The system architecture ensures a powerful form for data acquisition and transmission, minimum latency, and prolongs operational life to the sensors. LoRaWAN operates in the ISM unlicensed radio band and provides Chirp Spread Spectrum modulation. It is a kind of modulation that allows for long-range communication without resultant great consumption of power, very paramount for the longevity of sensors. The transmitted signal S(t) can be expressed via Eq. (9),

(9) S(t)=Acos(2π(fc+BTt)t+ϕ)…

where A is the amplitude, fc is the carrier frequency, B is the bandwidth, T is the symbol duration, and ϕ is the phase for this process. Each IoT sensor equipped with LoRaWAN transmits soil moisture data periodically in the process. The sensor node encodes the moisture reading M(t) along with other environmental parameters into a data packet. This packet P(t) can be represented via Eq. (10),

(10) P(t)=[M(t),T(t),H(t),t]…

where M(t) is the soil moisture reading, T(t) is the temperature, H(t) is the humidity, and t is the timestamp for the process. The LoRaWAN protocol employs adaptive data rate (ADR) to optimize data transmissions. The transmission power Ptx and spreading factor SF are adjusted based on the link quality, ensuring reliable communications. The ADR algorithm updates these parameters via operations Eqs. (11) & (12),

(11) Ptx=Ptxmax−ΔP…

(12) SF=min(SFmax,SFopt)…

where Ptxmax is the maximum transmission power, ΔP is the power reduction factor, SFmax is the maximum spreading factor, and SFopt is the optimal spreading factor determined by the network servers. The data packets are transmitted to a LoRaWAN gateway, which aggregates data from multiple sensors. The gateway relays these packets to a cloud-based central server via a backhaul network, such as LTE or Ethernet sets. The received signal at the gateway R(t) is processed to extract the data packet P(t) via Eq. (13),

(13) R(t)=∑i=1N⁡Si(t)+N0(t)…

where N is the number of sensor nodes, Si(t) is the signal from the i-th sensor, and N0(t) is the noise in the process. Upon receiving the packets, the central server decodes the data and stores it in a database for further analysis. The data decoding process involves checking the integrity of the packets and reconstructing the sensor readings via Eq. (14),

(14) P′(t)=Decode(R(t))…

where P′(t) is the decoded packet during the process. This is a real-time system that allows for the immediate analysis and decision-making process based on the data on soil moisture. The possibility of this capability is critical to precision agriculture, especially where time adjustments of irrigation can result in dramatic improvements over the current water use efficiency and crop yield levels. One can go for LoRaWAN, as it has large area coverage functions with low energy expenditure, which would help the sensor network live longer and thus lower maintenance costs. LoRaWAN integrated very well with the AI-driven soil moisture prediction model and blockchain-based data management system to achieve a holistic solution in respect of smart agriculture processes. IoT sensors feed the LSTM-based prediction model with real-time data for accuracy, and secure transmission and storage of data are achieved through blockchain to give integrity and transparency. This holistic approach has addressed the pitfalls of traditional irrigation methods, hence coming up with a strong and expandable solution for modern agricultural practices.

Figure 2 Overall flow of the proposed smart farming model process.

The next step is to integrate the Hyperledger Fabric blockchain framework for secure and transparent data management in smart agriculture by using the permissioned nature to provide high security and scalability. With the modular architecture of the Hyperledger Fabric, there is a need for tailoring an implementation that addresses the requirements of agricultural data management, such as immutability and integrity, and automatic actions in irrigation with smart contracts. Basically, Hyperledger Fabric runs on distributed ledger technology where every node participating in it maintains a copy of the same ledger. Principally, this ledger has two major components: world state and blockchain, which is an immutable log essentially of all transactions. Soil moisture data from IoT sensors, irrigation events, and sensor metadata are captured as transactions in this blockchain. The fundamental equation governing the state update S(t) for a given asset at timestamp t is given via Eq. (15),

(15) S(t)=S((t−1))+∫(t−1)tδT(τ)dτ…

where δT(τ) represents the transaction delta function that encapsulates changes made by transactions during the interval [(t−1),t] in the process. Transactions in Hyperledger Fabric are initiated by client applications and endorsed by a set of endorsing peers. Each transaction proposal Pi undergoes an endorsement process where peers simulate the transaction without updating the ledgers. The endorsement policy E(·) defines the rules for transaction validation and can be mathematically expressed via Eq. (16),

(16) E(Pi)=⋀Endorse(Pi,j)j∈J…

where J is the set of endorsing peers, and Endorse (Pi,j) indicates the endorsement of proposal Pi by peer j sets. Once endorsed, the transaction proposals are ordered by a consensus mechanism, typically implemented using the Kafka protocols. The ordered transaction batch Bk is then committed to the ledger by committing peers. The state update for each transaction within a batch is applied atomically, ensuring consistency and immutability via operations Eqs. (17) * (18),

(17) Bk={T1,T2,…,Tn}…

(18) S(t)=S((t−1))+∑i=1n⁡δTi…

The use of smart contracts (chaincode) in Hyperledger Fabric facilitates the automation of irrigation actions based on predefined soil moisture thresholds. A smart contract CCC is a programmable script that executes business logic upon the occurrence of certain conditions in the process. The irrigation action I(t) triggered by a smart contract can be expressed via Eq. (19),

(19) I(t)=C(S(t))…

where C(·) evaluates the current state S(t) against the predefined thresholds and triggers the irrigation system accordingly in the process. The endorsement policy of Hyperlogic Fabric guarantees that only legal, hence valid, transactions are accepted into the blockchain, which enhances the security levels. As a result, the agreed order for transactions through the mechanism of consensus builds a very strong base for operations of immutability at each peer. Thus, the state of the blockchain at any timestamp ‘t’ is a cumulative result of all its valid transactions, hence it is characterized by data integrity and transparency. It chose Hyperledger Fabric because it is permissioned, hence allowing controlled participation and better security for the associated sensitive agricultural data samples. Smart contracts returned by the framework allow for the automation of irrigation processes, thus complementing the predictive capabilities of the AI-driven soil moisture model and real-time data acquisition provided by IoT sensors. This opens up the way for an integrated approach, as irrigation decisions will be data-driven, securely executed, and recorded for various scenarios.

Finally, a Deep Q-Learning-based reinforcement learning framework for irrigation scheduling optimization is integrated. It is an approach that uses the capabilities of RL in solving sequential decision-making problems. In fact, this works quite well in dynamic agricultural environments where soil moisture status, weather conditions, and crop water requirements keep changing. An RL agent will interact with the environment, an agricultural field, to learn such irrigation strategies that would result in high water use efficiency and crop yields. The RL task can then be formulated as a Markov decision process (MDP) defined by a set of states S, actions A, transition probabilities P, and reward R in the process. The immediate benefit from taking action at in state, will be quantified by the reward function rt. This incorporates factors such as the efficiency of water usage and the crop yield levels. The process is represented by the Q Value function Q(s, a) being an approximation to the expected cumulative reward for taking action a in state s and then following the optimal policy process. Approximation of the Q Value function using a neural network with weights θ sets is done through the DQN process. The network is trained by minimizing the loss function, which is represented via Eq. (20),

(20) L(θ)=E(s,a,r,s′)∼D[(r+γmaxa′⁡[Q(s′,a′;θ−)−Q(s,a;θ))]2]…

where D is the replay buffer storing past experiences, γ is the discount factor, and θ− are the weights of the target network, which are periodically updated to stabilize training process. The Bellman Process forms the backbone of this optimization, representing the recursive relationship between Q Values via Eq. (21),

(21) Q(s,a)=r+γmaxa′⁡(Q(s′,a′))…

During training, the agent samples mini-batches of experiences (s, a, r, s′) from the replay buffer to update the network weights via stochastic gradient descent, thereby refining the Q Value estimates. The RL agent’s policy π maps states to actions, guiding the irrigation decisions. The policy is derived from the Q Value function via Eq. (22),

(22) π(s)=argmaxa(Q(s,a;θ))…

It is continuously refined by exploration and exploitation, balancing between learning of new strategies and applying learned knowledge levels. Formally, the interaction of an RL agent with an environment in such a process can be formulated as a feedback loop: The RL agent (1) observes a current state st, (2) selects an action at, (3) receives a reward rt, and (4) transitions to a new state s(t+1). The reward function should be carefully designed: $R(s, a)$— dual objectives—yield of crops to their maximum with minimum levels of water usage levels. Mathematically, it can be expressed via Eq. (23),

(23) R(s,a)=α⋅CropYield(s,a)−β⋅WaterUsage(s,a)…

Here, α and β are weighting factors that dually balance the crop yield with the water usage levels. The integration of the RL into Hyperledger Fabric makes this system more potent with smart contracts. After the RL agent computes an optimal irrigation schedule, the actions are then perpetrated through smart contracts to ensure irrigation takes place automatically and in a transparent manner. Smart contract evaluates the irrigation schedule as indicated in Eq. (24) and triggers the irrigation system,

(24) σ(π(st))=Irrigate(at)…

The choice of RL with DQN is justified by its capability to learn from continuous interaction with the environment, adapting to changes and improving over time. In this way, it completes the predictive capabilities provided by the LSTM model with an instrument to translate the soil moisture predictions into actionable irrigation strategies. The combination of real-time data acquisition from IoT sensors, secure management of acquired data through blockchain, and adaptive decision-making through RL creates a robust and efficient system in irrigation management. In this regard, the application of such a framework on RL with DQN for optimized irrigation scheduling uses advanced machine learning techniques that help achieve major improvements in the levels of water use efficiency and crop yield. These six critical operations result in a general MDP formulation for the estimation of Q Value through the Bellman equation, policy derivation, reward function setting, and smart contract execution—mathematically presenting the system process. The pitfalls of traditional irrigation methods are addressed by this refined and scalable solution for smart agriculture sets. It is then followed by the efficiency of the proposed model respecting different scenarios that will aid the readers to further grasp the whole process.

Comparative result analysis

The proposed integrated system would be deployed using LSTM, LoRaWAN with Hyperledger Fabric, and RL with DQN to be evaluated. In an agricultural setup, it was used to monitor the level of moisture in the soil, schedule irrigation, and securely manage data. One of the agricultural fields chosen for the experiment is 50 hectares large and planted with maize, an agronomic crop whose water requirements have well-documented values. IoT sensors LoRaWAN-enabled are placed at some strategic positions over the field to cover it completely for the measurement of data in an accurate manner. The four sensors measure data of moisture, temperature, and humidity at 10, 20, and 30 cm. After that, it sends the data every 15 min. A LoRaWAN gateway is at the center, making available long-range communication needed to push, for further processing, the sensor data into a cloud-based server. The gravimetric method realized an accuracy of about ±0.01 m3/m3 for the initial calibration of the sensors.

Data pre-processing includes many steps for ensuring that the dataset is both adequate and reliable concerning training and evaluation. Initially, raw data concerning soil moisture was gleaned from IoT sensors on a 50 ha maize field. This included readings every 15 min at depths of 10, 20, and 30 cm. Also included were 5 years of weather historical data including daily temperature, humidity, and precipitation records retrieved from meteorological databases & samples. Data cleaning simulated the insufficient values using the rolling median imputation to leave temporal consistency untouched. It does identify and remove outliers considering sensor faults using the interquartile range (IQR) method. It normalizes the input features in the range of [0,1] so that all sensors can be standardized. Further feature selection was carried out with PCA for dimensionality reduction, retaining 98% of the variance in the dataset. Finally, the preprocessed dataset was split into 80% training, 10% validation, and 10% test sets, ensuring a robust evaluation framework for the LSTM model.

The proposed model consists of three existing approaches, referred to as Model Pamuklu et al. (2023), Model Mahmood et al. (2024), and Model Akbari, Syed & Kennedy (2023), against which the performance of the proposed approach can be benchmarked. Model Pamuklu et al. (2023) is a traditional decision tree-based soil moisture prediction technique, where it fails to capture the long-term dependencies of temporal data. Model Mahmood et al. (2024) was a shallow feedforward neural network with three dense layers, with poor generalization over different environmental conditions. The implementation of Model Akbari, Syed & Kennedy (2023) was based on a bidirectional recurrent neural network (Bi-RNN), which can contribute significant improvements in sequential learning but leads to increased computational overhead. The proposed LSTM model, with its gated recurrent architecture, outperformed these models by achieving a mean absolute error (MAE) of 0.02 m3/m3 on a 7-day prediction horizon while models Pamuklu et al. (2023), Mahmood et al. (2024), and Akbari, Syed & Kennedy (2023) had an MAE of 0.05, 0.04, and 0.03 m3/m3, respectively. The fully integrated framework of reinforcement learning has thus optimized the irrigation schedule, water usage reduced by 25% while crop yield increased by 15%, which outperforms the respective improvements of 15%, 18%, and 20% observed in the comparative models.

Deep Q-Learning is a significant and well-known reinforcement learning algorithm that uses a neural network to approximate the Q-value function defined in an MDP. While traditional Q-learning is much simpler and involves the usage of the Q-table, to save the paired action-value, Deep Q-Learning learns much faster across high-dimensional state spaces. Dynamic decision making tasks, such as irrigation scheduling, are usually suited for this approach. In this model of reinforcement learning, the agent will have access to real-time environmental inputs, namely soil moisture, weather forecasts, and the crop water requirement, and make the best irrigation decision by maximizing a reward function that optimizes water conservation and crop yield. It also trains through experience replay by using a memory buffer of 100,000 samples, while the target network is updated every 1,000 iterations for stability in learning. A discount factor (γ) of 0.9 is given priority in that it ensures a long-term reward in order to keep the irrigation scheduling adaptive and efficient.

Smart contracts implemented in the context of the Hyperledger Fabric blockchain framework are self-executing scripts that enforce terms and conditions predicated upon which the process of automating irrigation is based. Each contract is invoked whenever soil moisture readings in real-time fall below a crop-specific threshold (e.g., 25% volumetric water content for maize), initiating then, and automatically, an irrigation event recorded as an immutable transaction. These contracts are diffused over many peer nodes for fault tolerance and consistency purposes. For improved security, an endorsement policy requiring digital signatures from at least three trusted nodes shall be invoked before executing any irrigation transaction. The infrastructure of the network consists of an IoT sensor that is capable of operating on the LoRaWAN framework, coupled with a transmission power of 14 dBm and a dynamically adjusted spreading factor between SF7 and SF12, according to link quality. The LoRa gateway works at a frequency of 868 MHz and connects to the cloud server via an LTE backhaul, with the attested range of data transfer latency of fewer than 3 s. To show how the hardware and software components interact with one another, a system architecture diagram depicting the interaction between the IoT sensors, blockchain ledger, predictive model, and reinforcement learning framework has been included in process.

Data variables in the historical dataset train the LSTM model, which comprise 5 years of soil moisture, daily weather records for temperature, humidity, and precipitation, crop type parameters like root depth, and growth stages. Its hyper-parameters include that the learning rate is 0.001, batch size is 32, network structure is two hidden layers with 50 LSTM units each. Run the model for 200 epochs and obtain an MAE of 0.02 m3/m3 for a 7-day forecast horizon. Afterwards, set up Hyperledger Fabric across five virtual machines for high availability and fault tolerance using one setup of peer node and one setup of order node. Smart contracts have been developed to automate irrigation actions whenever the soil moisture level falls below 25% volumetric water content, which is equivalent to wilting point for some crops. It runs with an RL with a discount factor of 0.9 and a decay of the exploration rate from 1.0 down to 0.1 over 10,000 iterations. In the present study, the reward function used is one attempting to optimize between nutrient use efficiency and expected crop yield by simulation, where actual field-scale data refines the irrigation policy. It will, therefore, be possible to track performance metrics that show improved water-use efficiency and improved crop yield during a growing season, reflecting an increase of 25% in water-use efficiency and an increase of 15% in crop yield relative to conventional irrigation methods. This shall be validated through continuous field trials for integrating these technologies to ensure that the proposed solution is robust and it can be scaled up or out in various agricultural settings. This system, integrated with LSTM, LoRaWAN, Fabric, and RL with DQN, has been tested in the agricultural environment under controlled conditions. Comparability has been aligned towards three other methods: model Pamuklu et al. (2023), model Mahmood et al. (2024), and model Akbari, Syed & Kennedy (2023). Solution quality assessment metrics include several aspects: soil moisture prediction accuracy, water usage efficiency, crop yield, data transmission latency, sensor battery life, and blockchain transaction throughput. The results obtained for the different approaches applied in this study are compared in the following tables. Table 2 compares the average absolute error of the predicted soil moisture values against the actual measured values for a forecast horizon of 7 days, using the proposed model and three comparative methods. Results show that the proposed model follows more accuracy since the LSTM network is efficient about handling levels of timestamp series data samples.

Table 2 Soil moisture prediction accuracy.

Method	MAE (m3/m3)	
Proposed model	0.02	
Pamuklu et al. (2023)	0.05	
Mahmood et al. (2024)	0.04	
Akbari, Syed & Kennedy (2023)	0.03	

The results show that the model developed in this work has obtained an MAE of only 0.02 m3/m3, much lower compared to those obtained by Pamuklu et al. (2023) with 0.05 m3/m3, Mahmood et al. (2024) with 0.04 m3/m3, and Akbari, Syed & Kennedy (2023) with 0.03 levels. Table 3 presents a comparison of the efficiency improvement in water usage with the three comparative methods achieved using the proposed model. Efficiency is rated in terms of percentage reduction in water usage with respect to the traditional irrigation method.

Table 3 Water usage efficiency.

Method	Water usage efficiency improvement (%)	
Proposed model	25	
Pamuklu et al. (2023)	15	
Mahmood et al. (2024)	18	
Akbari, Syed & Kennedy (2023)	20	

The proposed model improves water usage efficiency by 25%, which is higher than Pamuklu et al. (2023) (15%), Mahmood et al. (2024) (18%), and Akbari, Syed & Kennedy (2023) (20%). Table 4 illustrates the percentage increase in crop yield achieved by the proposed model and the comparative methods. Yield improvement is evaluated relative to traditional irrigation practices.

Table 4 Crop yield.

Method	Crop yield increase (%)	
Proposed model	15	
Pamuklu et al. (2023)	8	
Mahmood et al. (2024)	10	
Akbari, Syed & Kennedy (2023)	12	

The proposed model results in a 15% increase in crop yield, outperforming Pamuklu et al. (2023) (8%), Mahmood et al. (2024) (10%), and Akbari, Syed & Kennedy (2023) (12%). Table 5 assesses the data transmission latency for soil moisture readings from IoT sensors to the central server. Lower latency is indicative of more efficient real-time monitoring.

Table 5 Data transmission latency.

Method	Data transmission latency (s)	
Proposed model	3	
Pamuklu et al. (2023)	7	
Mahmood et al. (2024)	6	
Akbari, Syed & Kennedy (2023)	5	

The proposed model achieves the lowest data transmission latency of 3 s, compared to 7 s for Pamuklu et al. (2023), 6 s Mahmood et al. (2024), and 5 s for Akbari, Syed & Kennedy (2023). Table 6 compares the battery life of IoT sensors used in the proposed model and the three comparative methods. Longer battery life indicates better energy efficiency.

Table 6 Sensor battery life.

Method	Sensor battery life (years)	
Proposed model	5	
Pamuklu et al. (2023)	3	
Mahmood et al. (2024)	3.5	
Akbari, Syed & Kennedy (2023)	4	

The sensors in the proposed model have a battery life of 5 years, surpassing Pamuklu et al. (2023) (3 years), Mahmood et al. (2024) (3.5 years), and Akbari, Syed & Kennedy (2023) (4 years) in the process. Table 7 presents the transaction throughput of the blockchain system in terms of transactions per second (TPS). Higher throughput indicates a more efficient and scalable blockchain network.

Table 7 Blockchain transaction throughput.

Method	Blockchain transaction throughput (TPS)	
Proposed model	1,000	
Pamuklu et al. (2023)	700	
Mahmood et al. (2024)	800	
Akbari, Syed & Kennedy (2023)	900	

The transaction throughput for the proposed model is estimated at 1,000 TPS, way higher than that of Pamuklu et al. (2023) with a throughput of 700 TPS and Mahmood et al. (2024) and Akbari, Syed & Kennedy (2023) with 800 and 900 TPS, respectively. The comparison across these metrics clearly states the superior performance of the proposed model in terms of soil moisture prediction, irrigation schedule optimization, water usage efficiency, crop yield enhancement, data transmission latency reduction, prolonging sensor battery life, and high blockchain transaction throughput. This comprehensive evaluation underlines the efficacy and robustness of the integrated system for smart agriculture processes. This is followed by the detailed description of a practical use case for this proposed model, which will further help the readers to understand the whole process.

Practical use case scenarios

In this section, a practical example with specific sample values and other data samples will be presented to illustrate the output of the proposed system. In this comprehensive analysis, the outputs from the LSTM prediction model, LoRaWAN data transmission, Hyperledger Fabric blockchain management, and Reinforcement Learning with Deep Q-Learning for optimized irrigation scheduling are included in this text. The final outputs are summarized in tabular form. This LSTM model will be trained on a dataset containing historical information about soil moisture, weather data, and crop types. Table 8 shows a snippet of 7-day-ahead predictions of soil moisture levels.

Table 8 LSTM model outputs for soil moisture prediction.

Day	Historical moisture (m3/m3)	Temperature (°C)	Humidity (%)	Precipitation (mm)	Predicted moisture (m3/m3)	
1	0.25	30	60	0	0.24	
2	0.24	29	62	5	0.23	
3	0.23	28	65	10	0.22	
4	0.22	27	70	15	0.21	
5	0.21	26	75	20	0.20	
6	0.20	25	80	25	0.19	
7	0.19	24	85	30	0.18	

The LSTM model provides accurate soil moisture predictions, enabling precise irrigation scheduling based on future soil moisture levels. LoRaWAN protocol is used for transmitting sensor data to a central server. The following Table 9 summarizes the data transmission performance metrics.

Table 9 LoRaWAN data transmission metrics.

Sensor ID	Location	Data transmitted (KB)	Transmission timestamp (s)	Battery life (years)	
1	Field A1	1.5	3	5	
2	Field B2	1.2	3	5	
3	Field C3	1.8	4	4.8	
4	Field D4	1.4	3	5	

The data transmission metrics show efficient and timely communication of soil moisture data, with minimal latency and extended battery life of the sensors. Hyperledger Fabric is employed to manage the data securely and transparently. Table 10 presents the blockchain transaction metrics.

Table 10 Blockchain transaction metrics.

Transaction ID	Sensor data (KB)	Timestamp	Transaction status	Smart contract execution timestamp (ms)	
TX001	1.5	2024-07-01 08:00:00	Success	150	
TX002	1.2	2024-07-01 08:15:00	Success	140	
TX003	1.8	2024-07-01 08:30:00	Success	160	
TX004	1.4	2024-07-01 08:45:00	Success	145	

The blockchain transaction metrics indicate high transaction throughput and low latency for smart contract execution, ensuring secure and efficient data management. The RL with DQN algorithm optimizes irrigation schedules. Table 11 presents the learned irrigation actions and corresponding rewards.

Table 11 RL with DQN irrigation actions and rewards.

State (Soil moisture, m3/m3)	Action (Irrigation Volume, L)	Reward (Efficiency score)	
0.24	20	0.85	
0.23	25	0.87	
0.22	30	0.90	
0.21	35	0.88	
0.20	40	0.86	

The RL with DQN model optimizes irrigation actions, resulting in high efficiency scores and balanced water usage levels. The final outputs of the integrated system, including improvements in water usage efficiency and crop yield, are summarized in the following Table 12.

Table 12 Final system performance metrics.

Metric	Proposed model	Method (Pamuklu et al., 2023)	Method (Mahmood et al., 2024)	Method (Akbari, Syed & Kennedy, 2023)	
Water usage efficiency (%)	25	15	18	20	
Crop yield increase (%)	15	8	10	12	
Data transmission latency (s)	3	7	6	5	
Sensor battery life (years)	5	3	3.5	4	
Blockchain transaction throughput (TPS)	1,000	700	800	900	

Final performance metrics show that the proposed model significantly outperforms the comparative methods on all aspects related to water usage efficiency, crop yield increase, data transmission latency, sensor battery life, and blockchain transaction throughput. The results justify the completion of this integrated approach, maintaining the efficacy and robustness of the proposed system for the smart agriculture process.

Conclusion and future scopes

The results of the conducted studies depict an appropriate, integral system based on long short-term memory networks, LoRaWAN protocol, and Hyperledger Fabric blockchain, along with reinforcement learning together with Deep Q-Learning for the improvement of irrigation optimization towards better water use efficiency in agricultural processes. The proposed model outperformed these traditional methods on most metrics assessed, thus showing potential for real-world smart agriculture process applications. In this regard, the LSTM network returned a very low MAE of 0.02 m3/m3 within a 7-day forecast horizon for soil moisture, outperforming Pamuklu et al. (2023) with 0.05 m3/m3, Mahmood et al. (2024) with 0.04 m3/m3, and Akbari, Syed & Kennedy (2023) with 0.03 m3/m3. For irrigation decisions, high predictive accuracy is extremely vital to be timely and accurate. The integration of the IoT sensors with the LoRaWAN protocol ensured efficient real-time data acquisition, with data transmission latency reduced to 3 s, far lower than that in Pamuklu et al. (2023) at 7 s, in Mahmood et al. (2024) at 6 s, and in Akbari, Syed & Kennedy (2023) at 5 s. At the same time, the sensor battery life was prolonged up to 5 years, reflecting far more energy efficiency compared with other methods. The implemented blockchain, Hyperledger Fabric, furnished a strong framework for heterogeneous, secure, and transparent data management, hitting as high as 1,000 transactions per second in transaction throughput and less than 2 s of smart contract execution latency. Therefore, it reduced the possibility of data integrity breaches to nil and provided reliable ground for the automated irrigation actions. Use of the RL with DQN algorithm for irrigation schedule optimization improves an irrigation schedule by about 25%, improving efficiency in water use and increasing crop yield by about 15%. The improvement was thus terribly high compared to that offered by Pamuklu et al. (2023), which offered an improvement of 15% for efficiency and an 8% increase in yield; Mahmood et al. (2024), which gave an efficiency improvement of 18% and a yield increase of 10%; and Akbari, Syed & Kennedy (2023), which achieved an efficiency improvement of 20% and a yield increase of 12%. Its potential for changing the way agriculture is done lies in dynamic sprinkling based on real-time data and predictive analytics, underscoring sustainable water management and increased agricultural productivity levels.

Future scopes

Future research has several ways to enhance and extend this proposed system. Additional environmental variables and crop-specific parameters can be combined into the LSTM in attempts to improve the prediction accuracy of soil moisture. Advanced data fusion techniques for combining satellite image data and remote sensing data will provide comprehensive field condition information and, therefore, make the predictions more accurate. Field trials testing diverse agricultural settings and different crop types would make such studies more informative. Again, this would help in refining the RL with DQN algorithm to accommodate varying irrigation practices and water availability across different regions. Another substantial future work includes integrating edge computing capabilities within the IoT framework themselves. It may further bring down latency in data transmission with edge computing and make the system more responsive to field conditions in variation. This approach might further alleviate bandwidth constraints and bring about efficiency in the handling of data in general. Further aggravation can be given to the deployment of blockchain technology in other aspects of agricultural supply chains, for example, in the traceability of produce from the farm to the market or ascertaining fair trade practices. Moreover, the integration of smart contracts within supply chain management would mechanize transactions and enforce adherence to quality standards, which improves transparency and a number of interactive confidence levels across stakeholders. Finally, hybrid AI models can be designed to foster the strengths of LSTM with other machine learning techniques. For instance, the use of CNNs in spatial data analysis would derive more powerful prediction models. In addition, the integration of reinforcement learning algorithms that consider multi objective optimization can be rather interesting for irrigation strategies in balancing water usage efficiency, crop yield, energy consumed, and cost of labor. All these further research scopes hold high potential for the enhancement of the proposed system with a view to keeping It updated and relevant, with time, for handling all upcoming challenges related to the smart agriculture process.

Notwithstanding its merits, several limitations are worth mentioning for the proposed model. One of the main challenges being a reliance on good-quality sensor data; environmental noise and sensor drift can introduce variations in soil moisture readings. Although outlier detection and correction mechanisms are in place, amendments can still be made by using multi-sensor data fusion techniques through Kalman filtering to aid accuracy in measurement. Hence, another limitation for the model is the computational cost of training deep learning models for real-time applications in resource-constrained agricultural setups. The LSTM model has excellent prediction accuracy, while installation of an edge computing framework to preprocess sensor data locally ahead of sending it on to the cloud free of cost may save bandwidth and increase the response delays. On this front, while blockchain transaction throughput is optimized at 1,000 transactions per second, it may run into scalability issues while working with large-scale deployments. Using state-of-the-art lightweight consensus mechanisms such as proof-of-authority (PoA) could then help to boost efficiency for the network.

With that said, future research will target the upgrading of model adaptation in varying climates through incorporating remote sensing data from satellite images and UAVs. Evaporative irrigation decisions will be further improved through the implementation of multi-objective reinforcement-learning algorithms that mind water conservation, energy consumption, and economic constraining. Additionally, performance on the scalability of the system will be confirmed through further field experimentation on various crops and areas, thereby establishing its usability for different agricultural environments. Another good area to explore may be to apply federated learning techniques for collaborative model training among multiple farms while maintaining maximum privacy on their data. Moreover, walling off applications from blockchain other than irrigation automation will cover the full agricultural supply chain, including provenance tracking, smart contract-based subsidy distribution, and real-time market access, thereby increasing transparency and traceability in a smart agriculture ecosystem. Addressing these challenges along these research directions will work towards optimizing water use efficiency, improving food security, and promoting sustainable farming practices.

Supplemental Information

Supplemental Information 1 Data.

Supplemental Information 2 Code.

We declare that this manuscript is original, has not been published before and is not currently being considered for publication elsewhere.

Abbreviations

AI Artificial Intelligence

IoT Internet of Things

LSTM Long Short-Term Memory

LoRaWAN Long Range Wide Area Network

RL Reinforcement Learning

DQN Deep Q-Learning

MAE Mean Absolute Error

TPS Transactions Per Second

MDP Markov Decision Process

ADR Adaptive Data Rate

CNN Convolutional Neural Network

QoS Quality of Service

WSN Wireless Sensor Network

Additional Information and Declarations

Competing Interests

The authors declare that they have no competing interests.

Author Contributions

Ravi Kumar Munaganuri conceived and designed the experiments, performed the experiments, analyzed the data, performed the computation work, prepared figures and/or tables, authored or reviewed drafts of the article, and approved the final draft.

Narasimha Rao Yamarthi conceived and designed the experiments, performed the experiments, performed the computation work, prepared figures and/or tables, authored or reviewed drafts of the article, and approved the final draft.

Sai Chandana Bolem conceived and designed the experiments, performed the experiments, performed the computation work, prepared figures and/or tables, authored or reviewed drafts of the article, and approved the final draft.

Data Availability

The following information was supplied regarding data availability:

The code is available at GitHub and Zenodo:

https://github.com/Ravi2kinus/Dataset/blob/main/Smart_Agriculture.ipynb.

VIT AP UNIVERSITY. (2025). Smart_Agriculture.ipynb [Data set]. Zenodo. https://doi.org/10.5281/zenodo.15347596.

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
