# Peer review of "Design of an improved graph-based model integrating LSTM, LoRaWAN, and blockchain for smart agriculture"

_PeerJ Computer Science, doi:10.7717/peerj-cs.2896_

## Round 0.1 · original submission · Minor Revisions

Dear authors,

Your paper has been reviewed by three reviewers who suggested revisions. Please revise the paper, clearly mark all changes, and write a cover letter with responses to reviewers point to point.

Reviewer 1 ·

Basic reporting

no comment

Experimental design

no comment

Validity of the findings

no comment

Additional comments

no comment

Reviewer 2 ·

Basic reporting

1. Simplify the Language:
- The paper should use clear and simple language. For example, the sentence in lines 316 to 320 is very long and hard to understand. The author should use simple tenses and shorter sentences to help readers follow the ideas easily.

2. Explain Abbreviations:
- When using abbreviations, the full term should be written first. For example, write "Long Short-Term Memory" before using "LSTM." This rule also applies to abbreviations like AI, IoT, and ML. Some abbreviations, such as LoRaWAN, DQN, and RL, are not explained at all and need a clear definition.

3. Shorten Section Titles:
- The titles of each section should be short. For example, change "In-depth Review of Existing Models" to a shorter title like "Related Work."

Experimental design

- The paper missing some data analysis , what the process that has been done in the data preprocessing.
- The author should explain more about what is the model that they compare with in the experiment . For example, what exactly is the Model [15], etc.

Validity of the findings

No comment

·

Basic reporting

In general, academic English is used, but there is room for improvement, particularly regarding sentence length and clarity. Some sentences are too long and complex, making comprehension more difficult. Shortening and simplifying the structure of the text would improve readability and clarity of presentation.
The paper provides a good introduction to the topic of digital agriculture. Relevant and up-to-date references are included.
The structure of the paper aligns with PeerJ journal standards and academic norms. The key sections are clearly separated and logically connected

Experimental design

The paper is well-structured and focuses on a highly relevant topic. Due to its interdisciplinary nature and broad readership, it would be beneficial to introduce definitions for some technical terms such as Deep Q-Learning and Smart Contracts. I recommend including technical details about the sensors and network equipment used. Additionally, a diagram illustrating the hardware components could be provided

Validity of the findings

The paper presents an innovative and integrated approach to utilizing AI, IoT, and Blockchain technologies in smart agriculture. The conclusions are valid, but it would be beneficial to include a discussion on limitations and potential challenges in implementation, as well as directions for future research. To enable replication of the experiment, technical details about the sensors and network equipment should be included.

Additional comments

The manuscript focuses on a highly relevant and innovative topic and is well-structured in accordance with PeerJ journal standards and academic norms. The research approach and results have the potential to be highly valuable to the broader academic community. I believe that the minor revisions suggested above will make the manuscript more accessible to a wider academic audience.

---

## Round 0.2 · accepted · Accept

Dear authors,

It is a distinct pleasure to inform you that reviewers recommended acceptance of your paper after the last round of review.

·

Basic reporting

The authors revised the manuscript in accordance with the suggestions.

Experimental design

The authors revised the manuscript in accordance with the suggestions.

Validity of the findings

The authors revised the manuscript in accordance with the suggestions.

Additional comments

Congratulations to the authors.